# Campylobacteriosis Agents in Meat Carcasses Collected from Two District Municipalities in the Eastern Cape Province, South Africa

**DOI:** 10.3390/foods9020203

**Published:** 2020-02-16

**Authors:** Aboi Igwaran, Anthony I. Okoh

**Affiliations:** 1SAMRC Microbial Water Quality Monitoring Centre, University of Fort Hare, Alice 5700, South Africa; 2Applied and Environmental Microbiology Research Group (AEMREG), Department of Biochemistry and Microbiology, University of Fort Hare, Private Bag X1314, Alice 5700, South Africa

**Keywords:** *Campylobacter* species, contamination, infection, meat, virulence

## Abstract

Raw meats are sometimes contaminated with *Campylobacter* species from animal faeces, and meats have repeatedly been implicated in foodborne infections. This study evaluated the prevalence, virulence genes, antimicrobial susceptibility patterns, and resistance gene determinants in *Campylobacter* species isolated from retailed meat carcasses. A total of 248 raw meat samples were collected from butcheries, supermarkets, and open markets; processed for enrichment in Bolton broth; and incubated at 42 °C for 48 h in 10% CO_2_. Thereafter, the broths were streaked on modified charcoal cefoperazone deoxycholate agar (mCCDA) plates and incubated at the same conditions and for the same amount of time. After incubation, colonies were isolated and confirmed by Polymerase chain reaction using specific oligonucleotide sequences used for the identification of the genus *Campylobacter*, species, and their virulence markers. The patterns of antimicrobial resistance profiles of the identified isolates were studied by disk diffusion method against 12 antibiotics, and relevant resistance genes were assessed by PCR. From culture, 845 presumptive *Campylobacter* isolates were obtained, of which 240 (28.4%) were identified as genus *Campylobacter*. These were then characterised into four species, of which *C. coli* had the highest prevalence rate (22.08%), followed by *C. jejuni* (16.66%) and *C. fetus* (3.73%). The virulence genes detected included *iam* (43.14%), *cadF* (37.25%), *cdtB* (23.53%), *flgR* (18.63%), and *flaA* (1.96%), and some of the isolates co-harboured two to four virulence genes. Of the 12 antibiotics tested, the highest phenotypic resistance displayed by *Campylobacter* isolates was against clindamycin (100%), and the lowest level of resistance was observed against imipenem (23.33%). The frequency of resistance genes detected included *catll* (91.78%), *tetA* (68.82%), *gyra* (61.76%), *ampC* (55%), *aac(3)-IIa (aacC2)^a^* (40.98%), *tetM* (38.71%), *ermB* (18.29%), *tetB* (12.90%), and *tetK* (2.15%). There is a high incidence of *Campylobacter* species in meat carcasses, suggesting these to be a reservoir of campylobacteriosis agents in this community, and as such, consumption of undercooked meats in this community is a potential health risk to consumers.

## 1. Introduction

In the last decade, there has been a global upsurge in the rate of *Campylobacter* infections [1,2], and *Campylobacter* has emerged as one of the most significant bacteria of public health importance [3]. Globally, *Campylobacter* infection is a significant zoonosis, considered to be the leading cause of bacterial foodborne infection [4]. This zoonotic infection is of great public health concern [5], with meats known as the major risk factor [6] due to consumption of undercooked poultry or red meats [7]. Worldwide, consumption of meats and their products is increasing, and this may be connected to meats’ high protein content and the recommendations of healthy nutrition [8]. Meat is one of the most nutritious food items and a regular component of the human diet. Moreover, countless products, such as burgers and sausages, are products from meat, and as a result of the central role that meats occupy in human nutrition, their hygienic value is very essential for public health, as consumption of poor-quality meats may cause infections [9]. Infections arising from consumption of contaminated undercooked meats or food remain a global threat to public health [10]. Foodborne disease is a major public health problem due to its increasing incidence worldwide [11,12,13] and its huge burden of morbidity and mortality caused by bacterial infections [14]. *Campylobacter* is among the regular foodborne bacteria pathogens that are responsible for most foodborne disease outbreaks [15,16,17]. 

Worldwide, *Campylobacter* is among the major pathogens that cause bacterial gastroenteritis [18]. Such bacteria are microaerophilic bacteria with respiratory-type metabolism [19], and several species are known to cause infections [20], with *C. coli*, *C. lari*, and *C. jejuni* being the most common species implicated in human infections [21]. Other species, such as *C. upsaliensis*, *C. concisus, C. fetus*, and *C. ureolyticus*, have also been reported to be implicated in human gastrointestinal infections and periodontitis [22]. The *Campylobacter* infectious dose is about 500 colony forming unit/g depending on the physical conditions of the individual or age [23], and the infection is caused by the virulence mechanisms that are involved in toxin production, flagellar motility, adhesion, and invasion of epithelial cells [24]. In general, the burden of *Campylobacter* foodborne disease remains significantly high across the world [25], and regular monitoring and examination of meats are necessary to maintain food safety standards [26]. 

Foodborne disease outbreak is defined as a food poisoning occurrence involving more than two persons epidemiologically connected to a common food source [27]. The shocking listeriosis outbreak that happened recently in South Africa, which took over 218 lives, highlights the importance of good food safety practices and food monitoring [28]. It is commonly accepted that the actual occurrence rate of foodborne disease is obviously higher than the documented data due to limited surveillance capacity and under-reporting, particularly in developing countries [29]. *Campylobacter* infection is an infection that is labelled “self-limiting”, which rarely requires antimicrobial treatment [30,31]. Nevertheless, antimicrobial treatment is needed in persisting or severe campylobacteriosis cases, immunocompromised patients, and cases of extragastrointestinal symptoms [32,33]. The antibiotics employed in the treatment of severe *Campylobacter* infections include azithromycin and erythromycin, ciprofloxacin, and tetracyclines [34]. Alternative drugs of choice for systemic campylobacteriosis treatment include gentamicin and ampicillin [35]. However, there are regular reports on the increasing rate of *Campylobacter* resistance to currently used antibiotics, including macrolides, quinolones, and fluoroquinolones, which represent a significant threat to public health that is of global concern [36]. In this paper, we report on the prevalence, virulence markers, and antimicrobial resistance of *Campylobacter* species in retailed fresh meat carcasses in two district municipalities in the Eastern Cape Province of South Africa as part of our larger study on reservoirs of antibiotic resistance in the environment. 

## 2. Materials and Methods

### 2.1. Ethical Clearance

Ethical clearance was applied for on behalf of the study and granted by the University of Fort Harr Alice, South Africa research ethics committee with certificate reference no. OKO021IGW01. 

### 2.2. Study Area

The samples were collected in Chris Hani and Amathole District Municipalities, Eastern Cape, South Africa, with geographical coordinates 31.8743° S, 26.7968° E and 32.5842° S, 27.3616° E, respectively, and Figure 1 is a map showing the study areas.

### 2.3. Sample Collection

A total number of 258 meat samples (mutton, chicken, turkey, beef, and pork) were purchased from different retail markets, open markets, and butcheries in different locations and towns in Chris Hani and Amathole District Municipalities, Eastern Cape, South Africa. All the meat samples were aseptically packed into separate sterile plastic bags to prevent cross contamination and were transported to the laboratory for analysis in a cooler box with ice packs within six hours of collection. 

### 2.4. Microbiological Analysis of the Meat Samples

The meat samples were analysed following ISO 10,272 guidelines for isolation and identification of *Campylobacter* species [37,38]. Briefly, 25 g portions of the meat samples were homogenised in 245 mL of buffered peptone water (M614-500G, (Vadhani, Mumbai, India)). Thereafter, 10 mL of the homogenate was added into 90 mL of Bolton selective enrichment broth (1.00068.0500 Merck), to which Bolton broth selective supplement (1.00079.0010 Merck) with 5% (*v*/*v*) defibrinated horse blood (JMS, Singapore) was added, and the resulting mixture was incubated at 42 °C for 48 h under microaerophilic conditions in 10% CO_2_ in an HF151UV CO_2_ incubator. After the 48 h incubation period, a loopful of the inoculum from the enriched broth was streaked on modified charcoal cefoperazone deoxycholate agar (mCCDA) plates containing antibiotic selective supplement (CCDA selective supplement 1.00071.0010) and incubated under the same conditions and for the same amount of time stated above. Thereafter, colonies suspected to be *Campylobacter* based on colony morphology were picked and re-streaked onto blood agar base plates supplemented with 7% (*v*/*v*) defibrinated horse blood, and the plates were incubated under the same conditions and for the same amount of time.

### 2.5. DNA Extraction

Template DNA for PCR assay was extracted following the process described by Sierra-Arguello et al. [39] with slight modification. Briefly, colonies isolated from the blood agar plates were grown in 5 mL of Tryptone Soya Broth (TSB) for 48 h at 42 °C under microaerobic conditions in a 10% CO_2_ incubator. After incubation, 1 mL of the broth was centrifuged at 12,800 rpm for 5 min, and the cells were suspended in 400 µL of sterile distilled water in sterile 1.5 mL Eppendorf tubes. The suspensions were boiled for 10 min at 100 °C in a heating block and allowed to cool, after which the suspensions were centrifuged at 12,800 rpm for 5 min and the supernatants were collected and stored at −20 °C until ready for use.

### 2.6. Molecular Identification of the Genus Campylobacter 

A 439 bp part of the 16S rRNA gene was amplified using primer CAM220 F-GGTGTAGGATGAGACTATATA and CAM659 R-TTCCATCTGCCTCTCCC as reported by Moreno et al. [40]. A singleplex PCR assay was carried out in a 25 µL reaction volume (5 µL of the DNA, 12.5 µL master mix, 2 µL of primer, and 5.5 µL of nuclear free water), and the PCR cycling conditions were set at initial denaturation (95 °C for 5 min), followed by 33-cycle (94 °C for 1 min, 58 °C for 1 min, and 72 °C for 2 min), and the final extension was set at 72 °C for 2 min. Verification of the amplified PCR products was carried out by resolving them in 1.5% agarose gel stained with ethidium bromide at 135 volts for 30 min, which was detected under a short-wavelength UV light source; *C. jejuni* ATCC 33.560 was used as the positive control.

### 2.7. Molecular Classification of Campylobacter Species 

PCR amplification was further carried out to delineate the isolates to the species level for the detection of *C. lari*, *C. fetus*, *C. jejuni*, and *C. coli*. The primer sets used for the detection of these species are as reported by Yamazaki-Matsune et al. [41]. 

### 2.8. Molecular Detection of Virulence Genes

The identified *Campylobacter* species were further screened for the presence of invasion genes (*ciaB* and *iam*), adherence genes (*flaA* and *cadF*), a toxin gene (*cdtB*) and flagella synthesis, and a modification gene (*flgR*). The primer sets for the detection of *cdtB*, *flaA* and *cadF* genes were used as reported by Modi et al. [24], *iam* gene [42], *ciaB* gene [43] and *flgR*) gene [44]. 

### 2.9. Phenotypic Determination of Antimicrobial Resistance

Patterns of antimicrobial resistance of the identified *Campylobacter* species isolated from different meat types were studied using the Kirby–Bauer disk diffusion method according to Clinical and Laboratory Standard Institute (CLSI) 45] guidelines. The isolates were tested against 12 antibiotics regularly used in human and veterinary practices, comprising nine antimicrobial families, including tetracycline/doxycycline (30 μg), tetracycline (30 μg); penicillins/ampicillin (10 μg); macrolids/azithromycin (15 µg), erythromycin (15 μg); aminoglycoside/gentamicin (10 μg); lincomycin/clindamycin (2 μg); phenicols/chloramphenicol (30 μg); fluoroquinolones/ciprofloxacin (5 μg), levofloxacin (5 µg); cephalosporin/ceftriaxone (30 μg); and carbapenems/imipenem (10 µg). Briefly, 50 µL of the glycerol stock was suspended in 5 mL of Tryptone Soy Broth and incubated at 42 °C for 48 h in 10% CO_2_ in a CO_2_ incubator. After incubation, the broths were suspended in sterile normal saline solution, followed by adjustment of turbidity to 0.5 McFarland standard. The solutions were evenly spread using sterile cotton swabs on Müller Hinton agar plates supplemented with 5% defibrinated horse blood. After drying, antibiotic discs were dispensed using a disc-dispensing apparatus, and the plates were incubated at 42 °C for 24 h in a 10% CO_2_ incubator. *C. jejuni* (ATCC 33560) and *C. fetus* (ATCC 27374) were used as reference strains. The inhibition zones for tetracycline, doxycycline, ciprofloxacin, and erythromycin were interpreted following CLSI [45] breakpoints for *Campylobacter*. As there are no breakpoints available for ampicillin, azithromycin, gentamicin, clindamycin, chloramphenicol, levofloxacin, ceftriaxone, and imipenem for *Campylobacter*, the breakpoints established by CLSI [45] for *Enterobacteriaceae* were used for the interpretation of results.

### 2.10. Multiple Antibiotic Resistance (MAR) Index

For the determination of the multiple antibiotic resistance (MAR) index, the formula MAR = x/y, stated by Krumperman [46], was adopted where x = is the number of antibiotics to which the test isolate showed resistance and y = is the total number of antibiotics to which the test isolate has been evaluated for susceptibility.

### 2.11. Genotypic Assessment of Antibiotic Resistance Genes

Molecular screening of resistance genes was carried out on important resistance genes by both simplex and multiplex PCR assays on the isolates, which showed phenotypic resistance to the test antibiotics. The primer sets used for the detection of *tetA*, *tetB*, *tetC* and *tetD* genes were used as reported by Ng et al. [47], *tetK* and *tetM* genes [48], *gyrA* gene [49], *ermB* gene [50], *catI* and *catII* genes [51], *(aac(3)-IIa (aacC2)^a^* gene [52] and *IMI*, *KPC*, *VIM* and *bla*_OXA_-48-lik genes [53]. Verification of the amplified PCR products was carried out as stated above.

## 3. Results

### 3.1. Molecular confirmation of Campylobacter species

In the effort to isolate and detect *Campylobacter* species in meat samples, including pork, mutton, mutton offals, beef, beef offals, turkey, chicken, and chicken offals; the samples were subjected to both traditional culture and PCR techniques. From culture, a total of 845 presumptive isolates were obtained, of which 28.40% (208/845) were identified as belonging to the genus *Campylobacter* by PCR assay, of which 32.5% (208/640) were obtained from retail markets, 15.17% (22/145) from butcheries, and 16.67% (10/60) from open markets. The detailed results of the number of isolates from various meat types are shown in Table 1, while Figure 2 shows a representative gel image of the PCR confirmed genus *Campylobacter*.

### 3.2. Molecular Characterisation of Campylobacter Species

The 240 isolates identified as belonging to the genus *Campylobacter* were further delineated into four *Campylobacter* species by PCR technique: 53 (22.08%) isolates were identified as *C. coli*, 40 (16.66%) as *C. jejuni*, and 9 (3.75%) as *C. fetus*, whereas *C. lari* was not detected. A summary of the numbers of *Campylobacter* species identified in the meat types is shown in Table 2, while Figure 3 and Figure 4 are representative gel electrophoresis images of PCR-confirmed *C. jejuni*, *C. coli*, and *C. fetus*, respectively.

### 3.3. Molecular Detection of Virulence Genes

Among the 102 isolates characterised as *C. coli*, *C. fetus*, and *C. jejuni*; the presence of six virulence genes (*iam*, *cdtB*, *ciaB*, *cadF*, *flgR*, and *flaA*) were assessed by PCR. The virulence genes detected included *iam* (43.14%), *cadF* (37.25%), *cdtB* (23.53%), *flgR* (18.63%), and *flaA* (1.96%), and the detailed results are shown in Table 3. Moreover, 26 (25.49%) isolates co-harboured two virulence genes, 9 (8.82%) isolates co-harboured three virulence genes, and three (3.94%) isolates co-harboured four different virulence genes, and the patterns of distribution of the virulence makers co-harboured in the isolates are shown in Table 4, while Figure 5 and Figure 6 are gel electrophoresis images of PCR-confirmed *cadF*, *iam*, *flgR*, *cdtB*, and *flaA* genes.

### 3.4. Antibiotic Phenotypic Characteristics of the Identified Campylobacter Species

A total of 240 PCR-confirmed genus *Campylobacter* isolates were profiled for their possible phenotypic resistance against 12 antibiotics belonging to nine antimicrobial families. Astonishingly, all the isolates displayed the highest phenotypic resistance against clindamycin (100%). The *Campylobacter* isolates recovered from meat carcasses also displayed high phenotypic resistance against ampicillin (97.08%), tetracycline (94.17%), doxycycline (93.75%), erythromycin (87.03%), azithromycin (84.58%), ceftriaxone (83.75%), ciprofloxacin (76.25%), chloramphenicol (71.67%), gentamicin (64.58%), and levofloxacin (54.58%), and the lowest level of resistance was observed against imipenem (23.33%). Table 5 shows the detailed phenotypic resistance patterns observed in the isolates tested against the antibiotics, while Figure 7 is the interpreted antimicrobial susceptibility result according to CLSI [45] guidelines.

### 3.5. Assessment of Resistance Determinants 

The resistance genes screened among the phenotypic resistant-*Campylobacter* isolates are listed in Appendix A, and the choice for the selection of these genes was centred on their high phenotypic resistance rates. Thus, 12 antibiotic resistance genes were screened for probable detection of resistance genes among the identified *Campylobacter* species and also to determine the pattern of occurrence of multiple resistance genes in the isolates. From the PCR results obtained, the order of the frequency level of the resistance genes detected was as follows: *catll* (91.78%), *tetA* (68.82%), *gyra* (61.76%), *ampC* (55%), *aac(3)-IIa (aacC2)^a^* (40.98%), *tetM* (38.71%), *ermB* (18.29%), *tetB* (12.90%), and *tetK* (2.15%). In contrast, the *IMI*, *KPC*, *VIM*, *bla*_OXA_-48-like, *catl*, *tetC*, *tetD*, and *tetK* genes were not detected. Figure 8 and Figure 9 are representative gel images of the amplified PCR products of the assessed antibiotic resistance genes. Similarly, the patterns of the level of detected multiple resistance genes are shown in Table 6.

## 4. Discussion

Reports on the prevalence, virulence marker, and antimicrobial resistance genes in *Campylobacter* isolates recovered from retailed meat samples are well documented in some parts of the world, but limited information is available in some provinces in South Africa, particularly in the Eastern Cape Province. Hence, this study aimed to address the prevalence and characterise the identified *Campylobacter* species, virulence genes, and resistance genes in *Campylobacter* isolates recovered from meat carcasses. *Campylobacter* species are major bacteria foodborne enteropathogens that are regularly spread to humans through the consumption of contaminated food including meats [54]. In this study, a high rate of *Campylobacter* was detected in 240 (28.40%) isolates from meat carcasses, and this gives valuable insight into the possible risks of foodborne infection to humans. Studies carried out in Italy by Stella et al. [55], in Malaysia [56], in China [57], in France [58], and in South Korea [59] also detected *Campylobacter* in meat samples with detection rates of 34.10%, 50.9%, 48.9%, 76%, and 31.67%, respectively, and our findings are in line with their reports. Other studies carried out in Spain by García-Sánchez et al. [60], in Pakistan by Nisar et al. [3], in Yangzhou, China by Zou et al. [61], and in Northern Poland by Andrzejewska et al. [62] also detected *Campylobacter* in meat samples, and these results are also akin with their reports. According to Seliwiorstow et al. [63], different sampling sources make an impact on the occurrence rates of *Campylobacter*, which indicate the risk factors associated with handling. The occurrence rates of *Campylobacter* detected in the different market sources are in the order of 32.5% (retail markets), 16.67% (open markets), and 15.17% (butcheries). The highest occurrence rates of the genus *Campylobacter* were also detected in isolates from mutton (44.4%), followed by beef (34%), turkey (31.3%), beef offals (31%), chicken offals (29%), chicken (27%), and pork (25.2%). The occurrence rate of *Campylobacter* in mutton samples was much higher than that in other meat types, and this result is in agreement with the report of Maktabi et al. [64]. 

In the present study, the high detection rate of *Campylobacter* was observed in isolates from beef carcasses, and this result also corroborates the reports of Kashoma et al. [65] and Premarathne et al. [66]. The high rates of *Campylobacter* detection in turkey samples observed in this study also support those reported by Noormohamed and Fakhr [67] in Oklahoma, USA (17%), and Szosland-Fałtyn et al. in Poland (47.37%) [68]. The 240 isolates identified belonging to the genus *Campylobacter* were then characterised into four species, of which 53 (22.08%) were identified as *C. coli*, 40 (16.66%) as *C. jejuni*, and 9 (3.73%) as *C. fetus*, whereas *C. lari* was not detected. Higher rates of *C. coli* and *C. jejuni* were detected than other *Campylobacter* species, and our findings corroborate the report of Hodges et al. [69], Ocejo et al. [70], Sulaiman et al. [71], and Vinueza-Burgos et al. [72]. In contrast, a low prevalence rate of *C. fetus* was detected, and this finding is also in agreement with the report of Sinulingga et al. [56]. *C. coli* and *C. jejuni* are known to cause infection in humans, but from the first report of *Campylobacter* infection to date, *Campylobacter* pathogenesis has not been clearly understood. Though, what has been clear about *Campylobacter* infections and has been proposed as virulence determinants includes *Campylobacter*’s invasive capability, adherence to intestinal mucosa, ability to produce toxins, and flagella-mediated motility [73]. Thus, the presence of these specific genes associated with *Campylobacter* invasion, adhesion, toxin production, and colonisation are all essential for the process of infection, and the mechanism by which they cause disease in humans is assumed to be multifactorial [74]. From the PCR results of the virulence gene assessed, most *Campylobacter* species were detected to harbour a high proportion of *cadF*, *flaA*, and *iam* genes responsible for colonisation, invasion, and adherence, and our results corroborate the report of Abu-Madi et al. [75]. The *iam* gene had the highest occurrence rate of 43.14% (Table 3) among the various virulence genes screened, and our results also corroborate the report of Redondo et al. [76]. 

Similarly, the high occurrence rate of the *cadF* (37.25%) gene and the lower prevalence of the *flaA* (1.96%) gene were detected in the *Campylobacter* isolates from meat samples, and these results are akin with the reports of Andrzejewska et al. [77] and Ripabelli et al. [78]. Our results also corroborated the report of Chukwu et al. [74], who also detected the *cadF* gene in *Campylobacter* strains isolated from water and paediatric stools. The *ciaB* gene was screened for but was not detected, and our findings are contrary to the reports of Melo et al. [79], Melo et al. [80], and Zhong et al. [81] who reported high detection rates of the *ciaB* gene in meats and in retailed food samples. Another virulence gene screened for was the *cdtB* gene, and the presence and expression of any of the *cdt* genes (*cdtA*, *cdtB*, and *cdtC*) are essential for the efficient activity of the CDT toxin. In this study, the *cdtB* gene was observed to be widespread among the *Campylobacter* species, with *C. coli* strains revealed to have a higher prevalence rate of *cdtB* than *C. jejuni* and *C. fetus* (Table 3), and our results are not in agreement with the reports of Wieczorek et al. [82] and Reddy and Zishiri [83], which reported a higher prevalence rate of the *cdtB* gene in *C. jejuni* than in the *C. coli* strain. Multiple virulence genes were also detected in both *C. coli* and *C. jejuni* with *C. coli* observed to harbour more virulence genes than *C. jejuni*, and these results are in contrast with the reports of Lim et al. [84] and Zhang et al. [85]. *Campylobacter* isolates have also been reported by Han et al. [86] to co-harbour more than three virulence genes, and our findings corroborate this report. Virulence genes in the *Campylobacter* genome are known to be implicated in human infection, and *Campylobacter* pathogenicity may be strengthened by the expression of a single virulence gene or multiple virulence genes that are enough to establish infection in the host [80]. 

The dissemination of virulence-associated genes in the identified *Campylobacter* strains isolated from meats showed a potential risk to humans and an impending risk of outbreak of *Campylobacter* infection if appropriate measures are not put into place. From the antimicrobial susceptibility testing result, a total of 78 antibiotic resistance profiles were generated among the 102 *Campylobacter* isolates characterised as *C. coli*, *C. jejuni*, and *C. fetus*, and 76.47% were resistant to more than two antimicrobial families and were classified as multi-drugs resistant strains (Table 5). The most common observed resistant profile was LEV-CRO-C-CIP-E-ATH-IMI-CD-T-GM-DXT-AP, which was common among *C. coli* and *C. jejuni* strains. The isolates displayed high phenotypic resistance to tetracycline (94.17%), erythromycin (87.03%), ampicillin (97.08%), and ciprofloxacin (76.25%). In Brazil, studies have shown that there is a high prevalence of quinolones (72.2%), tetracycline (43%), erythromycin (38.9%), and ampicillin (26.9%) resistance in *Campylobacter* in circulation [87,88]. In Bolivia, Argentina, Chile, and Peru, many *Campylobacter* strains are resistant to quinolones (47–78%) as well as tetracycline (40.8–65.9% in Argentina and Bolivia), erythromycin (58.6% to 61.4% in Bolivia and Chile), and ampicillin (47.2% in Argentina) [89]. The value of the MAR index is 0.2, and the MAR index is a good risk assessment tool used to distinguish high- and low-risk areas where antibiotics are overused [90]. The MAR indices of the isolates were calculated, and 77.83% were found to have a MAR index greater than 0.2, while 17 isolates had MAR indices of 1.0 (Table 5). A MAR index value of greater than 0.2 indicates a high-risk source of contamination where antibiotics are often used, and based on our results, most isolates had MAR indices of greater than 0.2, confirming that there is high selective pressure and high antibiotic use in these areas.

Because the effectiveness of antibiotic resistance might be compromised in the treatment of infections, antimicrobial resistance genes were screened. Multiple antibiotic resistance genes were highly detected in most of the isolates (Table 6), and this is in agreement with the multiple phenotypic resistance profiles displayed by the isolates (Table 5). Multiple resistance genes in the *Campylobacter* isolates were detected, and our results are in agreement with the reports of Chukwu et al. [74] and Wieczorek et al. [91]. There was a high rate of detection of resistance genes in chloramphenicol (*catll* (91.78%)), tetracycline (*tetA* (68.82%)), ciprofloxacin (*gyra* (61.76%)), ampicillin (*ampC* (55%)), gentamycin *(aac(3)-IIa (aacC2)^a^* (40.98%)), and tetracycline (*tetM* (38.71%) in the *Campylobacter* isolates. In this study, the high detection rate of resistance genes in tetracycline/ciprofloxacin-*Campylobacter* isolates was responsible for its phenotypic resistance, and this is in line with the report of Nguyen et al. [92]. A high rate of the *gyra* gene was also detected in the *Campylobacter* isolates recovered from meat carcasses, and our results are also in agreement with the report of Du et al. [93]. The results from the study of Rahimi et al. [94] show that *Campylobacter* isolates recovered from meat carcasses were all susceptible to chloramphenicol and gentamycin, and our results are contrary to their report. The implications in antimicrobial resistance show a strong connection between the use of antibiotics in animal production, veterinary medicine, and antibiotic-resistant *Campylobacter* isolates in humans [95], although the majority of the antibiotics used in the treatment of bacterial infections in humans are also used in animals. Nevertheless, a One Health approach to addressing the issue of antibiotic resistance and the spread of antibiotic-resistant bacteria includes plans to maintain and carry on with the efficiency of current antibiotics by abolishing their inappropriate use and by preventing the spread of infection [96]. Another One Health approach in the prevention of human campylobacteriosis and the solution to the spreading of antibiotic-resistant bacteria is to improve animal, environmental, and human health with key components such as access to safe food, clean water, and hygiene [97]. In addition, recent One Health approach in the prevention and spread of antimicrobial resistance have focused primarily on the reduction of the use of antibiotics in food animals [98].

## 5. Conclusions

We investigated the prevalence rate, characterisation, distribution patterns of virulence genes, antibiotic susceptibility patterns, and antibiotic resistance genes in *Campylobacter* species isolated from meat carcasses obtained from butcheries, open markets, and supermarkets. High prevalence rates of the genus *Campylobacter* and virulence markers were detected in meat samples obtained in Chris Hani and Amathole District Municipalities, Eastern Cape, South Africa, and PCR assay is one of the appropriate methods for the detection and characterisation of virulence genes and resistance genes in bacteria species. The majority of the isolates showed resistance to the test antibiotics, and multi-resistant isolates were also observed. In conclusion, there should be a continues surveillance of the presence of these pathogens and antibiotic resistance genes in *Campylobacter* isolates, and awareness of the impending risks associated with the consumption of undercooked, contaminated meats should be increased.

## Figures and Tables

**Figure 1 foods-09-00203-f001:**
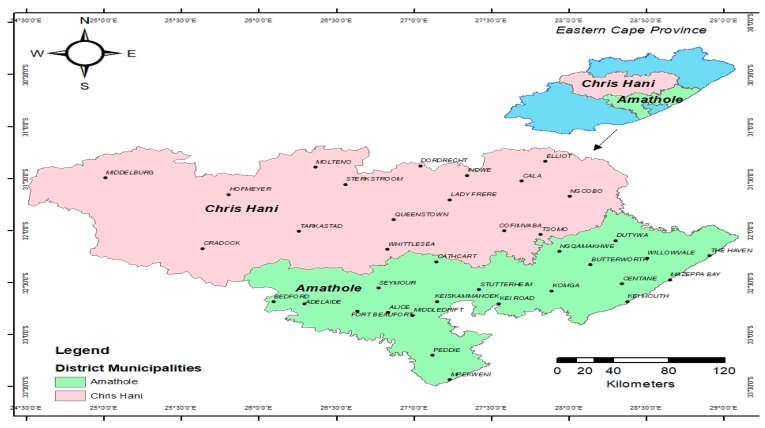
Map of Chris Hani and Amathole District Municipalities in the Eastern Cape Province, South Africa.

**Figure 2 foods-09-00203-f002:**
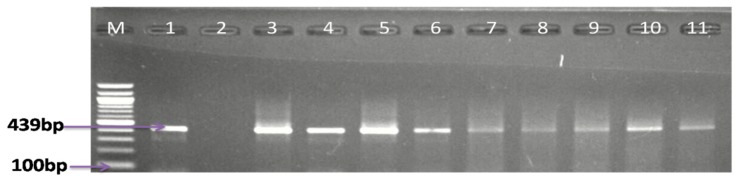
Agarose gel electrophoresis image of PCR-confirmed genus *Campylobacter*. Lane M: molecular marker (100 bp); lane 1: positive control (*C. jejuni* ATCC 33560); lane 2: negative control; and lanes 3−11: positive *Campylobacter* isolates (439 bp).

**Figure 3 foods-09-00203-f003:**
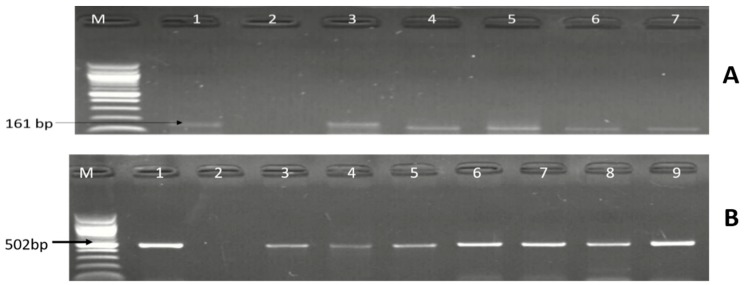
Gel (**A**) is a representative gel electrophoresis image of PCR-confirmed *C. jejuni*. Lane M: DNA ladder (100 bp); lane 1: positive control (*C. jejuni* ATCC 33560); lane 2: negative control; lanes 3–7: positive *C. jejuni* isolates (161 bp); and (**B**) is a representative gel image of PCR-confirmed *C. coli*. Lane M: DNA ladder (100 bp); lane 1: positive control (*C. coli* ATCC 33559); lane 2: negative control; and lane 3–9: positive *C. coli* isolates (502 bp).

**Figure 4 foods-09-00203-f004:**
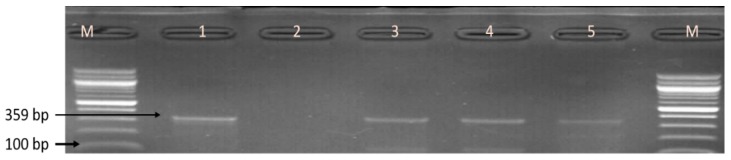
Gel picture of some PCR-identified *C. fetus*. Lane M: molecular weight marker (100 bp); lane 1: positive control (*C. fetus* ATCC 27374); lane 2: negative control; lanes 3–5: positive *C. fetus* isolates (359 bp).

**Figure 5 foods-09-00203-f005:**
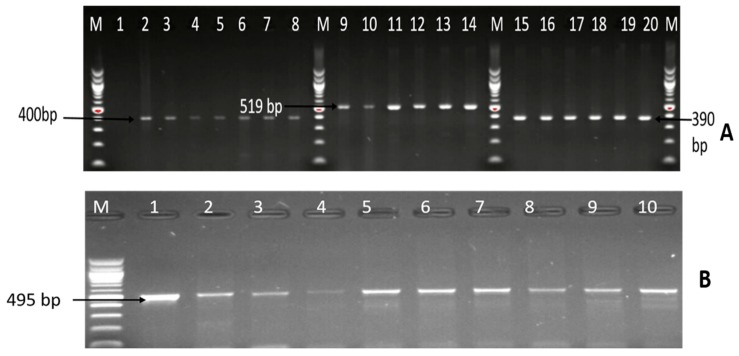
Image (**A**) is a representative gel picture of the PCR-confirmed *cadF*, *iam*, and *flgR* genes. Lane M: molecular marker (100 bp); lane 1: negative control; lanes 2–8: positive *Campylobacter* isolates that harboured the *cadF* gene (400 bp); lanes 9–14: positive *Campylobacter* isolates that harboured the *iam* gene (519 bp); lanes 15–20: positive *Campylobacter* isolates that harboured the *flgR* gene (390 bp); and (**B**) is a gel image of the PCR-confirmed *cdtB* gene. Lane M: DNA ladder (100bp); lanes 1–10: positive *Campylobacter* isolates that harboured the *cdtB* gene (495 bp).

**Figure 6 foods-09-00203-f006:**
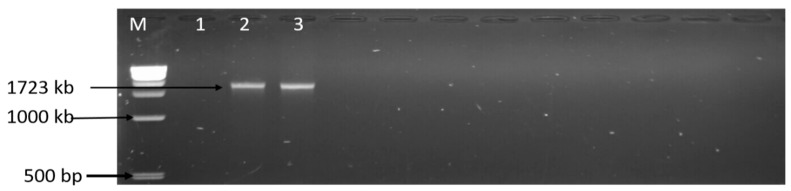
Gel image of the PCR-confirmed *flaA* gene. Lane M: DNA ladder (1 kb); lane 1: negative control; lanes 2–3: positive *Campylobacter* isolates that harboured the *flaA* gene (1723 kb).

**Figure 7 foods-09-00203-f007:**
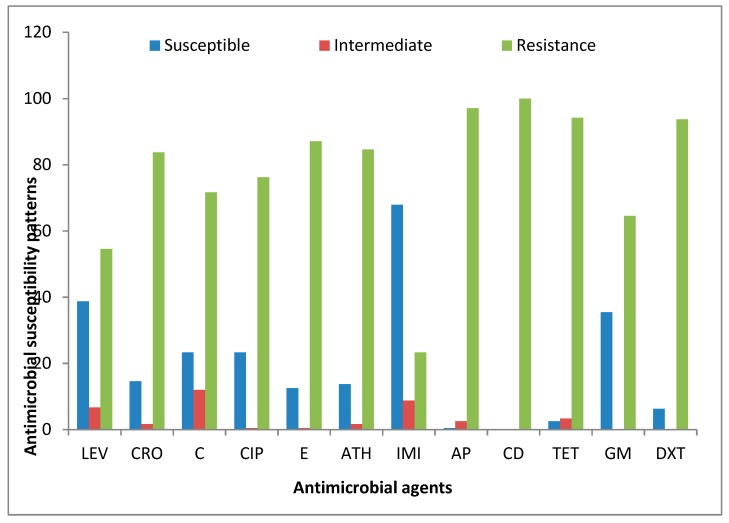
Antimicrobial susceptibility patterns of *Campylobacter* isolates recovered from meat carcasses sold in Eastern Cape, South Africa. Levofloxacin (LEV), ciprofloxacin (CIP), azithromycin (ATH), imipenem (IMI), ampicillin (AP), clindamycin (CD), tetracycline (TET), ceftriaxone (CRO), chloramphenicol (C), erythromycin (E), gentamicin (GM), and doxycycline (DXT).

**Figure 8 foods-09-00203-f008:**
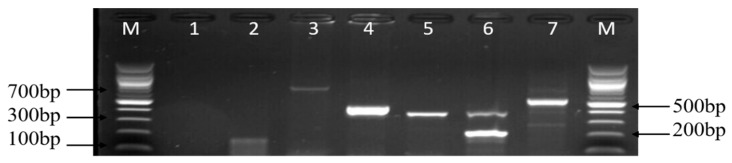
A representative gel image of various amplified antimicrobial resistance genes of *Campylobacter* isolates. Lanes M: DNA ladder (100 bp); lane 1: negative control; lane 2: *tetM* gene (158 bp); lane 3: *aac(3)-IIa (aacC2)^a^* gene (740 bp); lane 4: *gyrA* gene (441 bp); lane 5: *ermB* gene (320 bp); lane 6: *tetA* gene (201 bp) and *tetB* gene (359 bp); and lane 7: *ampC* gene (530 bp).

**Figure 9 foods-09-00203-f009:**
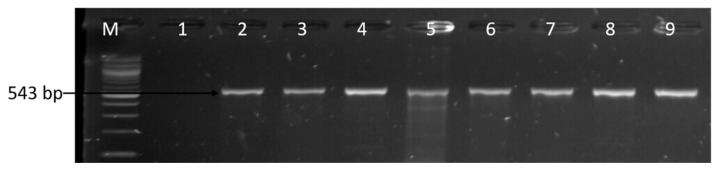
Gel image of the PCR-confirmed *catll* gene. Lane M: DNA ladder (100 bp); lane 1: negative control; and lanes 2–9, *Campylobacter* isolates that harboured the *catll* gene (543 bp).

**Table 1 foods-09-00203-t001:** Number of *Campylobacter* isolates identified in various meat types.

Meat Types	No. of Samples	No. of Presumptive *Campylobacter* Isolates	No. of Isolates Identified as Genus *Campylobacter*
Turkey	11	16	5 (31.25%)
Pork	35	131	33 (25.19%)
Mutton	22	9	4 (44.44%)
Mutton offals (heart)	2	6	0
Beef	27	89	30 (33.71%)
Beef offals (intestine, kidney, and liver)	31	126	39 (30.95%)
Chicken	68	300	81 (27%)
Chicken offals (liver, gizzard, and heart)	52	165	48 (29.09%)

**Table 2 foods-09-00203-t002:** Summary of *Campylobacter* species identified in the meat types.

Meat Typologies	*C. jejuni*	*C. coli*	*C. lari*	*C. fetus*	No. of *Campylobacter* Species Detected in the Meat Types
Pork	4	9	0	3	16
Beef	3	0	0	2	5
Beef offals	10	15	0	1	26
Chicken	9	16	0	2	27
Chicken offals	14	5	0	1	20
Mutton	0	0	0	0	0
Mutton offals	0	3	0	0	3
Turkey	0	5	0	0	5
Total	40	53	0	9	102

**Table 3 foods-09-00203-t003:** Percentage distribution pattern of detected virulence genes in the identified *Campylobacter* species.

Virulence Genes Screened	*Campylobacter* Species		
	*C. jejuni* (%)	*C. coli* (%)	*C. fetus* (%)
*iam*	7 (6.86)	35 (34.31)	2 (1.96)
*cadF*	4 (3.92)	34 (33.33)	0
*flgR*	11 (10.78)	8 (7.84)	0
*cdtB*	6 (5.88)	17 (16.67)	1(0.98)
*flaA*	0	2 (1.96)	0
*ciaB*	0	0	0

**Table 4 foods-09-00203-t004:** Patterns of occurrence of multiple virulence genes in the identified species.

	Pattern of Multiple Virulence Genes	Number of *Campylobacter* Species				Total Number
		*C. coli*	*C. fetus*	*C. jejuni*	-	
1	*iam*, *cadF*	16	-	1	-	17
2	*iam*, *flgR*	2	-	-	-	2
3	*iam*, *cdtB*	1	-	-	-	1
4	*cadF*, *flgR*	-	-	1	-	1
5	*cadF*, *cdtB*	4	-	-	-	4
6	*flgR*, *cdtB*	1	-	-	-	1
7	*iam*, *cadF, flaA*	1	-	-	-	1
8	*iam*, *cadF*, *cdtB*	7	-	-	-	7
9	*cadF*, *cdtB*, *flgR*	1	-	-	-	1
10	*iam*, *cadF*, *flgR*, *cdtB*	2	-	1	-	3

**Table 5 foods-09-00203-t005:** Antibiotic resistance patterns of *Campylobacter* isolates isolated from meat carcasses.

No	Multiple Antimicrobial Resistance Profile	No of Isolates			Total	MAR Index
		*C. coli*	*C. jejuni*	*C. fetus*		
1	C-CD-AP	1	-	-	1	0.25
2	C-E-ATH-CD-AP	1	-	-	1	0.42
3	CRO-C-E-CD-AP	-	-	1	1	0.42
4	E-ATH-CD-T-DXT-AP	-	1	-	1	0.5
5	LEV-CRO-CIP-ATH-CD-AP	-	-	1	1	0.5
6	CRO-C-CIP-E-ATH-CD-AP	-	1	-	1	0.58
7	CRO-E-ATH-CD-T-DXT-AP	-	1	-	1	0.58
8	CRO-C-CIP-E-ATH-CD-AP	-	1	-	1	0.58
9	C-E-ATH-CD-T-DXT-AP	1	-	-	1	0.58
10	LEV-CRO-C-CIP-E-ATH-CD-AP	1	-	-	1	0.67
11	CRO-C-E-ATH-T-GM-DXT-AP	1	-	-	1	0.67
12	CRO-C-E-ATH-CD-T-DXT-AP	1	1	-	2	0.67
13	CRO-CIP-E-ATH-CD-T-DXT-AP	1	1	-	2	0.67
14	CRO-E-ATH-CD-T-GM-DXT-AP	-	1	-	1	0.75
15	CRO-C-CIP-E-ATH-CD-T-DXT-AP	-	-	1	1	0.75
16	CRO-CIP-E-ATH-IMI-CD-T-DXT-AP	-	-	1	1	0.75
17	LEV-CRO-CIP-E-ATH-CD-T-DXT-AP	-	1	-	1	0.75
18	CRO-C-CIP-ATH-CD-T-GM-DXT-AP	1	-	-	1	0.75
19	LEV-CRO-C-CIP-E-ATH-CD-T-DXT	1	-	-	1	0.75
20	CRO-C-E-ATH-CD-T-GM-DXT-AP		2	1	3	0.75
21	LEV-CRO-C-CIP-E-ATH-CD-T-AP	-	-	1	1	0.75
41	LEV-CRO-C-CIP-E-ATH-CD-GM-DXT-AP	-	1	-	1	0.83
22	LEV-CRO-C-CIP-E-ATH-CD-T-DXT-AP	6	8	-	14	0.83
23	LEV-CRO-CIP-E-ATH-CD-T-GM-DXT-AP	1	-	-	1	0.83
24	CRO-C-CIP-E-ATH-CD-T-GM-DXT-AP	-	1	-	1	0.83
25	LEV-CRO-CIP-E-ATH-CD-T-GM-DXT-AP	-	1	-	1	0.83
26	LEV-CRO-C-CIP-E-ATH-CD-T-GM-DXT-AP	8	8	-	16	0.92
27	CRO-C-CIP-E-ATH-IMI-CD-T-GM-DXT-AP	-	1	-	1	0.92
28	LEV-CRO-C-CIP-E-ATH-IMI-CD-GM-DXT-AP	-	1	-	1	0.92
29	LEV-CRO-C-CIP-E-ATH-IMI-CD-T-GM-DXT-AP	14	3	-	17	1

**Table 6 foods-09-00203-t006:** Distribution and pattern of multiple antibiotic resistance of *Campylobacter* species isolated from meat carcasses.

No	Distribution Pattern of Antibiotic Resistance Determinants	No of Isolates			Total
		*C. coli*	*C. jejuni*	*C. fetus*	
1	*tetA*, *catII*	1	-	3	4
2	*tetM*, *catII*	2	-	-	2
3	*ampC*, *catII*	1	-	-	1
4	*catII*, *ermB*	1	-	-	1
5	*tetA*, *ampC*	-	-	1	1
6	*tetA*, *catII*, *gyra*	-	-	1	1
7	*tetA, tetB*, *ampC*	-	1	-	1
8	*tetA*, *ampC*, *catII*	2	2	-	4
9	*tetA*, *catII*, *ermB*	1	-	-	1
10	*tetM*, *ampC*, *catII*	1	-	-	1
11	*tetA*, *tetK*, *aac(3)-IIa (aacC2)^a^*	-	1	-	1
12	*ampC*, *catII*, *aac(3)-IIa (aacC2)^a^*	-	1	-	1
13	*tetM*, *catII*, *aac(3)-IIa (aacC2)^a^*	2	-	-	2
14	*tetA*, *ampC*, *catII*, *gyrA*	2	1	1	4
15	*tetA*, *ampC*, *catII*, *ermB*	-	2	-	2
16	*tetA*, *tetM*, *catII*, *ermB*	2	-	-	2
17	*tetA*, *tetB*, *ampC*, *gyrA*	-	1	-	1
18	*tetA*, *tetB*, *ampC*, *catII*	1	1	-	2
19	*tetA*, *tetM*, *tetK*, *catII*	-	1	-	1
20	*tetA*, *catII*, *ermB*, *gyrA*	-	1	-	1
21	*tetA*, *tetB*, *catII*, *ermB*	1	-	-	1
22	*tetA*, *tetM*, *ampC*, *catII*	1	-	-	1
23	*tetA*, *tetM*, *ampC*, *gyrA*	1	-	-	1
24	*tetM*, *ampC*, *catII*, *gyrA*	1	-	-	1
25	*tetM*, *catII*, *gyrA*, *aac(3)-IIa (aacC2)^a^*	1	-	-	1
26	*tetM*, *ampC*, *gyrA*, *aac(3)-IIa (aacC2)^a^*	1	-	-	1
27	*tetA*, *ampC*, *catII*, *aac(3)-IIa (aacC2)^a^*	1	3	-	4
28	*tetA*, *tetM*, *ampC*, *catII*, *gyrA*	3	1	-	4
29	*tetA*, *tetB*, *tetM*, *catII*, *gyrA*	1	-	-	1
30	*tetA*, *tetB*, *ampC*, *catII*, *gyrA*	-	3	-	3
31	*tetA*, *ampC*, *catII*, *ermB*, *gyrA*	-	2	-	2
32	*tetA*, *tetM*, *ampC*, *catII*, *aac(3)-IIa (aacC2)^a^*	1	-	-	1
33	*tetA*, *tetM*, *catII*, *gyrA*, *aac(3)-IIa (aacC2)^a^*	1	-	-	1
34	*tetM*, *ampC*, *catII*, *gyrA*, *aac(3)-IIa (aacC2)^a^*	1	-	-	1
35	*tetA*, *ampC*, *catII*, *gyrA*, *aac(3)-IIa (aacC2)^a^*	1	2	-	3
36	*tetA*, *tetM*, *ampC*, *catII*, *ermB, gyrA*	2	1	-	3
37	*tetA*, *tetB*, *tetM*, *ampC*, *catII*, *ermB*	1	-	-	1
38	*tetA*, *tetM*, *ampC*, *catII*, *gyrA*, *aac(3)-IIa (aacC2)^a^*	-	1	-	1
39	*tetA*, *tetB*, *tetK*, *ampC*, *ermB*, *aac(3)-IIa (aacC2)^a^*	-	1	-	1
40	*tetA*, *tetB*, *ampC*, *catII*, *gyrA*, *aac(3)-IIa (aacC2)^a^*	-	1	-	1
41	*tetA*, *tetM*, *ampC*, *catII*, *gyrA*, *aac(3)-IIa (aacC2)^a^*	2	-	-	2
42	*tetA*, *tetM*, *ampC*, *ermB*, *gyrA*, *aac(3)-IIa (aacC2)^a^*	-	1	-	1
42	*tetA*, *tetM*, *ampC*, *catII*, *gyrA*, *aac(3)-IIa (aacC2)^a^*	1	1	-	2
43	*tetA*, *tetM*, *ampC*, *catII*, *ermB*, *gyrA*, *aac(3)-IIa (aacC2)^a^*	-	1	-	1
44	*tetA*, *tetB*, *tetM*, *ampC*, *catII*, *gyrA*, *aac(3)-IIa (aacC2)^a^*	1	-	-	1

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
