# Peer review of "Campylobacteriosis Agents in Meat Carcasses Collected from Two District Municipalities in the Eastern Cape Province, South Africa"

_foods, 2020, doi:10.3390/foods9020203_

Round 1

Reviewer 1 Report

Line  16: According ISO 10272 temperature shall be 41.5 °C instead of the stated 42 °C;

Line 100: Norm reference number is with no comma;

Line 105: Growth temperature in ISO 10272 is reported 41.5 °C not 42 °C (see also  lines 116, 150, 155...and where applied);

Line 107: "was" instead of "were";

Line 112: "same" instead of "sane";

Line 131: No comma in ATCC number;

Line 177: "of" should be used instead of "from";

Line 178-179: Not clear the percentage value and also in Table 4, they do not correspond;

Line 183-186: Figure 2 shows bands not well resolved, it is difficult to discriminate MW, should be improved;

Line 190: C. fetus percentage is 3.75% not 3.73% as reported, verify all calculation;

Line 196: Figure 3 Resolution of bands in gels are not properly clear;

Line 229: Figure 6 It is not reported the explanation of what in lane 3;

Line 262: Figure 8 the resolution of bands, lane 2 it is not clear, in lane 6 there are two bands that must be explained, in lane 7 explain why it is reported "AMPC"; in Figure 8 and 9 positive control is not included;

Line 296: It should be better using "our findings" instead of "our finding" as well as "our results" instead of "our result": other lines should be considered (302, 304, 332, 361 and verify whole paper);

Line 342: At least two families of antibiotics were used;

Line 370: It should be added from which geographycal location the strains where isolated;

Line 373: The sentence "High...." is not clear, it should be reconsidered in english;

Table 1 and 2 should be reconsidered and graphycally adjusted, e.g. sign "°" is referred to "C" and not to numbers,

When using percentage if two significative number after comma are considered this rule applies to all data (e.g. Line 182, Table 4; etc.), this because it is not clear if percentage data are rounded

Author Response

Reviewer 1 comments and responses to the comments

Reviewer 2 Report

The authors could expand the reach of the manuscript by introducing the one health concept and improving the discussion of the current study taking this into account .

The term "antimicrobial" is preferable to "antibiotic".

Section 5. Conclusions could be expanded to include the concepts of multiresistant bacterial strains and one health. Furthermore, the authors discuss the prevalence of Campylobacter species according to meat origin (butcheries, open markets and supermarkets) and meat type (mutton, chicken, turkey, beef, and pork).

Minor comments:

Table 6-replace "Iam" by "iam"

Figure 7-update legend-number of isolates. Campylobacter isolates and not Campylobacter species.

Supplementary Materials: Supplementary Tables 1, 2 and 3.

Author Response

Reviewer  2 comments and responses to comments

Round 2

Reviewer 2 Report

The authors failed to answer some of the questions that I raised:

-Please change "antibiotic" to "antimicrobial" and not the opposite.

-The One Health concept has not been properly addressed and discussed. There are several very recent papers on the One Health approach to prevention, treatment, and control of campylobacteriosis.

One example:

A One Health approach to prevention, treatment, and control of campylobacteriosis

Francesca Schiaffino;James Platts-Mills;Margaret Kosek;

Curr Opin Infect Dis. 2019 Oct;32(5):453-460. doi: 10.1097/QCO.0000000000000570.

...but there are several others.

Author Response

Attached document is the responses to the reviewer comments
